# Antidiabetic Treatment before Hospitalization and Admission Parameters in Patients with Type 2 Diabetes, Obesity, and SARS-CoV-2 Viral Infection

**DOI:** 10.3390/jpm13030392

**Published:** 2023-02-23

**Authors:** Patricia-Andrada Reștea, Mariana Mureșan, Adrian Voicu, Tunde Jurca, Annamaria Pallag, Eleonora Marian, Laura Grațiela Vicaș, Ionuț I. Jeican, Carmen-Bianca Crivii

**Affiliations:** 1Department of Preclinical Discipline, Doctoral School of Biomedical Science, Faculty of Medicine and Pharmacy, University of Oradea, 410087 Oradea, Romania; 2Department of Preclinical Discipline, Faculty of Medicine and Pharmacy, University of Oradea, 410087 Oradea, Romania; 3Department II, Faculty of Pharmacy, “Victor Babes” University of Medicine and Pharmacy, 300041 Timisoara, Romania; 4Department of Pharmacy, Faculty of Medicine and Pharmacy, University of Oradea, 410087 Oradea, Romania; 5Department of Morphological Sciences, “Iuliu Hațieganu” University of Medicine and Pharmacy, 400347 Cluj-Napoca, Romania

**Keywords:** SARS-CoV-2, diabetes, obesity, cytokine storm, admission parameters, COVID-19, insulin therapy

## Abstract

Background: SARS-CoV-2 viral infection is a current and important topic for patients with comorbidities of type 2 diabetes and obesity, associated with increased risk of mortality and morbidity. This study aims to analyze, compare and describe admission parameters in patients with type 2 diabetes, obesity, and SARS-CoV-2 infection based on whether they received insulin therapy before hospital admission. Methods: Our study enrolled patients diagnosed with type 2 diabetes, obesity, and SARS-CoV-2 viral infection, 81 patients without insulin treatment before hospital admission, and 81 patients with insulin at “Gavril Curteanu” Municipal Clinical Hospital of Oradea, Romania, between August 2020 and March 2022. RT-PCR/rapid antigen tests were used for detecting SARS-CoV-2 viral infection. Results: The severe form of COVID-19 was found in 66% of all patients (65% in the group without insulin and 67% in the group with insulin). Oxygen saturation at the time of hospital admission was greater or equal to 90% in 62% of all patients. The most associated comorbidities we founded in this study were: hypertension in 75% of all patients (grade two hypertension 63% in the group without insulin and 64% in the group with insulin), ischemic heart disease in 35% of patients (25% in the group without insulin and 44% in the group with insulin, *n* = 0.008), heart failure in 9.3% of all patients (8.6% in the group without insulin and 10% in the group with insulin). CRP and procalcitonin are increased in both groups at hospital admission, with a slightly higher trend in the group with insulin therapy before hospital admission. We found that 56% of patients in the group with insulin treatment were with uncontrolled diabetes on admission. Only 10% of patients required a change in antidiabetic treatment with insulin therapy at discharge. In our study, 89% of all patients did not require short-term home oxygen therapy at discharge. Conclusions: Antidiabetic therapy taken before hospital admission did not protect patients against cytokine storm in COVID-19, but is very important in the pathophysiological stage of comorbidities. Paraclinical parameters at hospitalization showed differences in correlation with oral antidiabetic treatment like metformin or insulin therapy. Changing the antidiabetic treatment for a small percentage of patients in the group who had not been receiving insulin therapy before discharge was necessary. It is necessary for future studies to see all changes involved in antidiabetic treatment in patients with diabetes type 2 and obesity after SARS-CoV2 viral infection and its long-term evolution.

## 1. Introduction

A major concern of the last two years, SARS-CoV-2 infections have prompted many studies, but there are still many unanswered questions about the association between insulin resistance, pancreatic beta cell damage, antidiabetic treatment, and infection with the novel coronavirus [1]. In COVID-19, a link was found between the dysfunction of the endothelium caused by associated comorbidities, vascular damage caused by SARS-CoV-2 viral infection, multiorgan failure, and mortality [2]. Infectious disorders are more likely to appear in a patient with diabetes, a substantial risk factor for coronavirus disease [3]. Obesity increases the risk of developing type 2 diabetes and respiratory infection because of its association with altering immune cells, insulin resistance, and low-grade inflammation [4]. Obesity and type 2 diabetes are both risk factors for SARS-CoV-2 viral infection and increase mortality risk [5]. The question is why some infected patients with type 2 diabetes and obesity had higher mortality than others with the same comorbidities [6]. It appears that there is a connection between the viral entry of SARS-COV2 and the use of insulin in the treatment of diabetic patients [7]. Apparently, infected patients who are at risk for diabetes or who have a history of type 2 diabetes and use insulin would require more frequent admission to intensive care [8]. Literature data show that insulin treatment would be linked with adverse outcomes in diabetic patients with COVID-19 [9,10,11]. Patients with COVID-19 have an aggressive inflammatory response, especially in severe forms, with the release of pro-inflammatory cytokines in cytokine storm; cytokine storm plays an important role in severe lung damage, multiple organ dysfunction syndromes, admission to the intensive care unit, and mortality [12]. The chronic inflammatory state in patients with type 2 diabetes and obesity facilitates the cytokine storm [13]. Type 2 diabetes and obesity can be associated with complications in patients with SARS-CoV2 viral infection [14,15,16].

The goal of this study was to assess and compare admission parameters in individuals with type 2 diabetes, obesity, and SARS-CoV-2 virus infection based on the type of antidiabetic medication they were using before hospitalization and how their antidiabetic medication changed after discharge.

## 2. Materials and Methods

### 2.1. Study Design

This study enrolled patients diagnosed with type 2 diabetes, obesity, and SARS-CoV-2 viral infection, 81 patients without insulin treatment before hospital admission, and 81 patients with insulin treatment before hospital admission at “Gavril Curteanu” Municipal Clinical Hospital of Oradea, Romania, between August 2020 and March 2022. RT-PCR/rapid antigen tests were used for detecting SARS-CoV-2 viral infection. This study was conducted in accordance with the Declaration of Helsinki. Before taking part in the study, each subject provided their informed, written agreement for inclusion.

The protocol of the study was approved by the Ethics Committee of “Gavril Curteanu” Municipal Clinical Hospital of Oradea (No. 32652/16.11.2020) and by the Ethics Commission of Oradea University (No. 5/A,21.09.2020).

*Inclusion criteria* were: type 2 diabetes, obesity, and COVID-19 confirmed by RT-PCR/rapid antigen test.

*Exclusion criteria* were: patients with type 1 diabetes, patients without type 2 diabetes, patients without obesity, and patients without COVID-19 confirmed by the RT-PCR/rapid antigen test.

This study included two groups of patients with type 2 diabetes, obesity, and SARS-CoV-2 confirmed infection, one group with oral antidiabetic treatment like metformin (81 patients) and another with insulin treatment (81 patients).

### 2.2. Data Collection

Data were collected from the electronic medical records and included the following parameters: demographic characteristics (age, gender, origin), COVID-19 severity, oxygen saturation on admission, other comorbidities, BMI, diabetic ketoacidosis on admission, uncontrolled diabetes on admission, transfer to intensive care unit, intubation and mechanical ventilation, short-term oxygen therapy for COVID-19 patient at discharge, insulin treatment on discharge and survival rate. The laboratory parameters collected were: inflammatory, hematological, biochemical, coagulation parameters and electrolytes.

### 2.3. Statistical Analysis

The statistical analysis was performed using the following software: R version 4.1.2 (R Development Core Team, R Foundation for Statistical Computing, Vienna, Austria), RStudio 2022.07.1 + 554 “Spotted Wakerobin” and JAMOVI version 1.8.3.0.

Since the data used are not part of a population with a normal distribution (*p*-value *<* 0.05 with the Shapiro–Wilk test), only non-parametric tests were used for the statistical analysis. For Table 1 and Table 2 we used: Median (IQR); *n* (%), Wilcoxon rank sum test, Fisher’s exact test, Pearson’s Chi-squared test. For Table 3 we used: Median (IQR); *n* (%), Wilcoxon rank sum test.

For survival analysis, we used the Kaplan–Meier curves and the log-rank test included in survival R package. [17]

## 3. Results

### 3.1. General Characteristics of Patients at Hospital Admission

Of the 162 patients, 81 (50%) without insulin therapy and 81 (50%) with insulin therapy were included in our study. The female patients prevailed (60%) compared to men (40%). The patients were aged between 39–91, with an increased percentage between 61–70 years old—38% of patients, followed by the range 71–80 years old—29%, with an average of 68 years. The patients from urban areas were 52%. BMI (median IQR) in the group without insulin was 32.90, characteristic of type 1 obesity, and in the group with insulin, BMI was 32.40 (median IQR of body mass index). The demographic analysis of the sample, according to insulin therapy before hospitalization, is summarized in Table 1.

### 3.2. The Severity of Disease, and Comorbidities at Hospitalization

The severe form of COVID-19 was found in 66% of all patients (65% in the group without insulin and 67% in the group with insulin). The most associated comorbidities were summarized in the Table 2 below.

### 3.3. Laboratory Parameters at Hospital Admission

During the laboratory tests performed at admission lymphocyte number was lower than normal laboratory value (68% of the patients in the group not receiving insulin and 53% of the patients in the group with insulin).

The following parameters were found to be with high values: C-reactive protein (93% of the patients not receiving insulin treatment and 95% of those on insulin therapy, *p*-value = 0.035), fibrinogen (86% of the patients in the group without insulin and 74% of those in the group with insulin), Creatinekinase (28% of the patients in the group without insulin and 41% of those in the group with insulin, *p* = 0.020). The results are summarized in Table 3 below.

## 4. Discussions

The severe form of COVID-19 was found in 66% of all patients (65% in the group without insulin and 67% in the group with insulin). Oxygen saturation in our study was greater or equal to 90% in 62% of all patients. Ischemic heart disease was founded in 25% of patients in group without insulin and in 44 % of patients in group with insulin therapy, with statistically significant the *p*-value 0.008.

The human pancreas can be a target of SARS-CoV-2 infection, and β-cell infection could contribute to the metabolic dysregulation/diabetes observed in patients with COVID-19 [18,19]. It is known that the novel coronavirus can influence the host cell metabolism and change the glycolysis process of the host [7,8,9,10]. Oral antidiabetic treatment in COVID-19 patients was associated with a decreased mortality rate, but insulin therapy increased the risk of adverse outcomes in type 2 diabetes [20,21]. In a randomized controlled study, metformin as an oral antidiabetic had a possible effect against severe forms of COVID-19, but high doses do not increase anti-inflammatory effects [22].

In a recent article, it was stated that metformin decreased viral entry of novel coronavirus into cells and protect against fibrosis, and had a protective effect in the microcirculation [23]. It seems that oral antidiabetic like metformin interferences with IL-6 cytokines levels involved in cytokines storm [23].

In a randomized study on metformin versus placebo in COVID-19, it was found that oral antidiabetic treatment had no clinical effects on ambulatory patients with diabetes and SARS-CoV-2 infection [24].

SARS-CoV-2 infection also interacts with epithelial cells of multiple organs and with the endothelium [25,26]. Evans et al. [27] have shown the crucial role of endotheliitis induced by SARS-CoV-2 and the fact that endothelium is an important source of inflammatory cytokines in the cytokine storm of COVID-19. Endothelial dysfunction is already present in comorbidities including diabetes, obesity, and cardiovascular disease, and the endothelial dysfunction brought on by SARS-CoV-2 infection is added to this [28]. There are theories proposed for the immunomodulatory and anti-inflammatory effects of some oral antidiabetics in diabetic patients with SARS-CoV-2 infection [29].

Clinical studies showed that diabetes in patients with COVID-19 disease is a risk factor for transfer to an intensive care unit and may be an independent factor for survival in SARS-CoV-2 viral infection [30,31,32]. In other studies, obesity in patients with SARS-CoV-2 viral infection was considered an independent risk factor for transfer to the intensive care unit [33,34]. For the prognosis, it is important how patients were treated during hospitalization and how they were treated before, but we analyzed the condition of patients at admission according to antidiabetic treatment before hospital admission because was necessary to identify from the beginning aspects that can show the severity of the disease. The patient’s condition at admission may be severe from home or worsen during admission. There are situations in which patients have been admitted with severe forms of COVID-19 in both groups, and our study showed the connection between the admission parameters and the treatment without insulin or with insulin at home. 

Most of the patients who were admitted to the hospital did not follow an antidiabetic diet, and during the COVID-19 pandemic, they no longer visited the doctor for blood glucose control and treatment modification due to pandemic restrictions or because of concern about contracting the SARS-CoV-2 virus. 

Laboratory parameters at hospitalization showed significant statistical differences in correlation with the type of antidiabetic treatment. CRP and procalcitonin are increased in both groups at hospital admission, with a slightly higher trend in the group with insulin therapy before hospital admission. Other studies have shown that a higher level of CRP, D-dimers, and decreased lymphocyte number are items included in cytokine storm in COVID-19 disease [35,36,37].

In other studies, metformin, sulfonylurea, and sodium-glucose Cotransporter-2 Inhibitors decreased mortality risk, but insulin therapy and gliptins are linked with a high risk of severe outcome and mortality in COVID-19 [38].

Literature data showed that in severe forms of COVID-19, oxygen therapy for the short term at home was recommended for hypoxemic patients, according to the protocols [39,40]. We found that the majority of all patients in this study did not require short-term oxygen therapy at home. Some patients in the group without insulin treatment needed a change in antidiabetic treatment after SARS-CoV-2 viral infection, requiring insulin therapy at discharge. The severity of COVID-19 in diabetic patients was primarily influenced by the pathophysiological stage of the disease according to literature data. The limitation of our study is given by the small number of patients, small groups, included in a single center, only admission parameters, case control study with the same number of patients in both groups. Our findings may be useful for the medical community adding information about antidiabetic treatment before hospitalization and admission parameters in patients with type 2 diabetes, obesity, and SARS-CoV-2 Viral Infection.

## 5. Conclusions

This research points out that antidiabetic therapy taken before hospital admission did not protect patients against cytokine storm in COVID-19, because the evolution of SARS-CoV-2 viral infection was primarily influenced by the pathophysiological stage of comorbidities. A small percentage of patients in the insulin-free group required a change in antidiabetic treatment at discharge in management with insulin therapy, and a high percentage of patients in both groups did not require short-term home oxygen therapy. Paraclinical parameters at hospitalization showed differences in correlation with the type of antidiabetic treatment, but, further, larger, multi-center studies are needed to explore the changes in antidiabetic therapy and the long-term evolution of type 2 diabetes and obesity following SARS-CoV-2 infection.

## Figures and Tables

**Table 1 jpm-13-00392-t001:** Patient Characteristics by Insulinotherapy at Admission.

Characteristic	*n*	Overall, *n* = 162 ^1^	No, *n* = 81	Yes, *n* = 81	*p*-Value
Age, Median (IQR)	162	68 (60–74)	68 (61–74)	66 (59–72)	0.390 ^2^
AgeGroup, *n* (%)	162				0.770 ^3^
<40		1 (0.6)	0 (0)	1 (1.2)	
>80		11 (6.8)	5 (6.2)	6 (7.4)	
40–50		15 (9.3)	8 (9.9)	7 (8.6)	
51–60		26 (16)	11 (14)	15 (19)	
61–70		62 (38)	30 (37)	32 (40)	
71–80		47 (29)	27 (33)	20 (25)	
Gender, *n* (%)	162				0.520 ^4^
man		64 (40)	30 (37)	34 (42)	
woman		98 (60)	51 (63)	47 (58)	
Origin, *n* (%)	162				0.160 ^4^
rural		77 (48)	34 (42)	43 (53)	
urban		85 (52)	47 (58)	38 (47)	
BMI, Median (IQR)	162	32.5 (30.9–34.5)	32.9 (31.2–34.3)	32.4 (30.9–34.6)	0.820 ^2^

^1^ Median (IQR); *n* (%); ^2^ Wilcoxon rank sum test; ^3^ Fisher’s exact test; ^4^ Pearson’s Chi-squared test.

**Table 2 jpm-13-00392-t002:** The severity of disease and comorbidities correlated to insulin administration on admission.

Characteristic	*n*	Overall, *n* = 162 ^1^	No, *n* = 81	Yes, *n* = 81	*p*-Value
Oxygen saturation, Median (IQR)	162	90.0 (88.0–92.0)	90.0 (89.0–92.0)	90.0 (88.0–91.0)	0.035 ^2^
Severity COVID19, *n* (%)	162				0.910 ^3^
mild		13 (8.0)	6 (7.4)	7 (8.6)	
moderate		42 (26)	22 (27)	20 (25)	
severe		107 (66)	53 (65)	54 (67)	
Day hospitalization, Median (IQR)	160	13.0 (10.8–14.0)	13.0 (10.0–14.0)	13.0 (11.0–15.0)	0.730 ^2^
Diabetic ketoacidosis, *n* (%)	162	9 (5.6)	2 (2.5)	7 (8.6)	0.170 ^4^
Hypertension type one, *n* (%)	160	6 (3.8)	4 (4.9)	2 (2.5)	0.680 ^4^
Hypertension type two, *n* (%)	160	102 (64)	49 (60)	53 (67)	0.390 ^3^
Hypertension type three, *n* (%)	160	4 (2.5)	3 (3.7)	1 (1.3)	0.620 ^4^
Ischemic heart, *n* (%)	162	56 (35)	20 (25)	36 (44)	0.008 ^3^
stroke, *n* (%)	162	12 (7.4)	4 (4.9)	8 (9.9)	0.230 ^3^
Heart failure, *n* (%)	162	15 (9.3)	8 (9.9)	7 (8.6)	0.790 ^3^
Intensive therapyunit, *n* (%)	162	32 (20)	16 (20)	16 (20)	>0.99 ^3^
imv, *n* (%)	162	27 (17)	12 (15)	15 (19)	0.530 ^3^
insulintherapy_discharge, *n* (%)	162				<0.001 ^3^
no		93 (57)	71 (88)	22 (27)	
Previously Insulin		58 (36)	0 (0)	58 (72)	
yes		11 (6.8)	10 (12)	1 (1.2)	

^1^ Median (IQR); *n* (%); ^2^ Wilcoxon rank sum test; ^3^ Pearson’s Chi-squared test; ^4^ Fisher’s exact test. imv means intubation and mechanical ventilation.

**Table 3 jpm-13-00392-t003:** Laboratory findings correlated to insulin administration on admission.

Characteristic	*n*	Scientific Unit [ ]	Overall, *n* = 162 ^1^	No, *n* = 81	Yes, *n* = 81	*p*-Value
Creatinekinase, Median (IQR)	153	U/L	105 (57–197)	82 (49–173)	121 (68–219)	0.020 ^2^
CRP, Median (IQR)	160	mg/L	97 (37–174)	75 (31–148)	116 (53–201)	0.035 ^2^
Blood Glucose, Median (IQR)	157	mg/dL	190 (145–277)	169 (131–220)	230 (163–321)	<0.001 ^2^
Fibrinogen, Median (IQR)	154	mg/dL	572 (455–665)	567 (463–645)	576 (451–672)	0.800 ^2^
LeukocytesNumber, Median (IQR)	161	*1000/uL	7.1 (5.5–9.9)	6.1 (4.8–8.4)	8.3 (6.6–11.3)	<0.001 ^2^
PlateletsNumber, Median (IQR)	161	*1000/uL	218 (169–299)	194 (163–260)	246 (184–344)	0.003 ^2^
NeutrophilsNumber, Median (IQR)	160	*1000/µL	5.8 (4.0–8.4)	4.7 (3.5–7.3)	6.8 (4.8–10.2)	<0.001 ^2^
LymphocytesNumber, Median (IQR)	160	*1000/µL	0.80 (0.60–1.20)	0.75 (0.60–1.02)	0.90 (0.60–1.40)	0.089 ^2^
MonocytesNumber, Median (IQR)	160	*1000/µL	0.50 (0.30–0.70)	0.50 (0.30–0.60)	0.50 (0.38–0.72)	0.033 ^2^

^1^ Median (IQR); *n* (%); ^2^ Wilcoxon rank sum test.

## Data Availability

Not applicable.

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
