# Peer review of "Antidiabetic Treatment before Hospitalization and Admission Parameters in Patients with Type 2 Diabetes, Obesity, and SARS-CoV-2 Viral Infection"

_jpm, 2023, doi:10.3390/jpm13030392_

Round 1

Reviewer 1 Report

This paper  is really very poorly written and suffers from global inconsistencies

1. It is unclear how were cases picked up

2. No data over metabolic control (glycaemia or HbA1c) or comorbidities of cases are given

3. Who is the control group in all analyses?

4. What is important for the prognosis is how patients were trerated during the hospitalization more than how they were prior to or after the hospitalization (see all reported by literature about the relation between insulin use and mortality after COVID)

5. tables and figures are unreadable

6 many details about clinical or radiological details, as well as on some narrative clinical or strumemental aspects (tiredness, TC aspects etc. are redundant. Many phrases in Introduction and Discussion  are not necessary and make the MS excessively wordy

All thios only to mention the main concerns.

My advice is to design and to evaluate all data after searching the cooperation and help from experts in COVID as well as in diabetes or epidemiology before preparing a manuscript 

Author Response

Our answer is in the attachment.

Reviewer 2 Report

The authors present a prospective cohort study of hospitalized obese T2DM patients with COVID infection comparing those with and without insulin treatment with respect to their clinical outcome.

The title of the paper is misleading, as the observational nature of the study cannot show any "effects" of insulin treatment, rather than simple associations between T2DM treatment and COVID outcome. It is apparent, that patients with insulin treatment often belong to the newly defined SIDD subtype (Ahlqvist et al. 2018, Zaharia et al. 2020), which itself has a higher risk for certain comorbidities but potentially a very limited responsiveness to other antidiabetic treatments.

The abstract is way too short and does neither contain any background information, nor the rationale of the study. Results are summarized too briefly and superficially.

Introduction: The overall rationale of the study is of interest, but is not shown transparently in the introduction. Several recent papers on the topic should be referenced:

https://pubmed.ncbi.nlm.nih.gov/36540082/

https://pubmed.ncbi.nlm.nih.gov/35680172/

https://pubmed.ncbi.nlm.nih.gov/35692410/

https://pubmed.ncbi.nlm.nih.gov/35256883/

https://pubmed.ncbi.nlm.nih.gov/34367067/

.....

Methods:

Please clarify, that the cohort consists of 162 patients in total, not 162 + 81 + 81, as it reads in the beginning of that section.

Please clarify the selection process for the study participants. Were they included consecutively without any patients declining participation? Please provide a flow chart on the cohort structure from pre-screening to inclusion.

The statistics section lacks information about the used tests for categorical variables.

Table 2 is poorly layouted. The phrase "at admission" is not necessary for each variable, as the title provides that information.

Tables and main text are very redundant; information about significant differences should be shortened in the text and linked to table content.

Table 1 needs BMI as additional variable for later adjustment.

Tables 2 ff. should be checked for the potential for statistical adjustment (age, sex, BMI).

Tables 4 and 6 lack scientific units.

All tables: p values should given consistently with three decimals.

Tables 5 and 6: The "moderate" group comprises both moderate and mild cases.

Data in tables 2-4 and 5-6 need correction for multiple testing.

Fig. 1-4 are visually hard to read.

Discussion: Needs to be amended in accordance to all previous comments and can only be fully evaluated after major revision.

Overall, the discussions section is poorly referenced for previous papers on the association of insulin treatment with poor COVID outcome. Most parts of the discussion merely summarize results, without bringing them into context with existing literature.

Author Response

Our answer is in the attachment.

Round 2

Reviewer 1 Report

I remain of the opinion that, although somewhat improved, this study has essential defects. For example, metabolic controlc is not assessed by HbA1c. My opinion is that the therapy during the hospitalization, in addition to the previous general conditions of these patients, is really important for their survival. The criterion of excluding patients only to equalize their number in analysis is, however, at least curious

Author Response

Our answer is in the attachment/

Reviewer 2 Report

The authors have revised their manuscript in accordance to the reviewer's suggestions. Many points have been addressed sufficiently.

Some minor points should be amended / clarified:

* Inclusion criterion "obesity" means mandatory BMI>30?

* BMI should be given with one decimal, only.

* BMI should be given in either table 1 or 2, but not both.

* Please clarify the p value for oxygen saturation; median and IQR appear to be very similar.

* The "characteristics column" of table 2 should be clearer; without underlines, but a clear term for each parameter.

* What is "hypertension one...two...three"?

* Why was ischemic heart disease placed between hypertension categories?

* Table 2 needs a legend explaining abbreviations.

* Significant differences (ischemic heart disease; oxygen saturation) should be mentioned in the text, too.

* 3.3: Several parameters are mentioned in the text, but not the tables. That's not transparent reporting.

* The terms "increase" or "decrease" imply a change over time, not a cross-sectional difference between groups. Please clarify.

* Table 3: Scientific units for blood cells are incorrect.

* Please use "blood glucose" or "plasma glucose" instead of "blood sugar".

* The discussion should start with a summary of the results and continue by putting them into context.

* The discussion mentions differences in fibrinogen levels. That's not correct.

* The discussion claims a significantly higher occurence of diabetic ketoacidosis in the insulin-treated group. This is not correct.

* The discussion states, that uncontrolled diabetes was overrepresented in the insulin-treated group. This statement is not justified by the data.

* The discussion and the conclusion claim an impact of insulin treatment on survival rates, which is disputed few lines between those statements. A significant impact on survival is not supported by your data.

* Strengths and limitations need to presented more extensively.

Author Response

Our answer is in the attachment.

Round 3

Reviewer 1 Report

I remain in the opinion that a background story of impaired glucose metabolism is yet worldwide a main predictor of mortality in patients with diabetes independently of Covid-19. I do not understand why HbA1c cannot be measured in patients at their Hospital admission: especially in this case. Removing data on survival obviously overcomes this problem, although the paper loses its initial meaning

No answer has been given to my question how patients were discarded for the study only to pair the numbers of cases and controls in  the analyses

Author Response

Our answer is in the attachment.

Reviewer 2 Report

The authors have revised their manuscript in accordance to the reviewer's suggestions.

Please assure using the correct term "creatinekinase" instead of "creatininekinase".

Author Response

Our answer is in the attachment.
